# Numerous cultivated and uncultivated viruses encode ribosomal proteins

Carolina M. Mizuno[1], Charlotte Guyomar[2], Simon Roux [3], Régis Lavigne[4], Francisco Rodriguez-Valera [5], Matthew B. Sullivan[6,7], Reynald Gillet[2], Patrick Forterre[1] & Mart Krupovic [1]

Viruses modulate ecosystems by directly altering host metabolisms through auxiliary metabolic genes. However, viral genomes are not known to encode the core components of translation machinery, such as ribosomal proteins (RPs). Here, using reference genomes and global-scale viral metagenomic datasets, we identify 14 different RPs across viral genomes arising from cultivated viral isolates and metagenome-assembled viruses. Viruses tend to encode dynamic RPs, easily exchangeable between ribosomes, suggesting these proteins can replace cellular versions in host ribosomes. Functional assays confirm that the two most common virus-encoded RPs, bS21 and bL12, are incorporated into 70S ribosomes when expressed in *Escherichia coli*. Ecological distribution of virus-encoded RPs suggests some level of ecosystem adaptations as aquatic viruses and viruses of animal-associated bacteria are enriched for different subsets of RPs. Finally, RP genes are under purifying selection and thus likely retained an important function after being horizontally transferred into virus genomes.

[1] Unité de Biologie Moléculaire du Gène chez les Extrêmophiles, Département de Microbiologie, Institut Pasteur, Paris 75015, France. [2] Univ Rennes, CNRS, IGDR (Institut de génétique et développement de Rennes) - UMR 6290, F-35000 Rennes, France. [3] Department of Energy Joint Genome Institute, Walnut Creek, CA 94598, USA. [4] Univ Rennes, Inserm, EHESP, Irset (Institut de recherche en santé environnement et travail)-UMR_S 1085, PROTIM, F-35000 Rennes, France. [5] Departamento de Producción Vegetal y Microbiología, Evolutionary Genomics Group, Universidad Miguel Hernandez, Alicante 03550, Spain. [6] Department of Microbiology, The Ohio State University, Columbus, OH 43210, USA. [7] Department of Civil, Environmental and Geodetic Engineering, The Ohio State University, Columbus, OH 43210, USA. These authors contributed equally: Carolina M. Mizuno and Charlotte Guyomar. Correspondence and requests for materials should be addressed to M.K. (email: krupovic@pasteur.fr)

During billions of years of co-evolution with their hosts, viruses have evolved numerous strategies to directly modulate metabolic pathways and subvert key cellular biosynthetic machineries, which ensure their successful propagation. For example, ocean viruses that infect cyanobacteria (cyanophages) commonly encode core photosynthetic reaction center proteins, which serve to maintain the complex photosynthetic machinery during infection[1,2]. These and other ocean viruses can similarly manipulate their host's ability to alter central carbon metabolism[3], uptake phosphate[4], and cycle nitrogen[5,6], and sulfur[7,8]—the fundamental building blocks of life. Complementarily, viruses employ a diverse array of host take-over strategies to (i) fight off host defenses by encoding anti-restriction-modification or anti-CRISPR genes[9,10], (ii) control transcription by encoding sigma factors or polymerases themselves[11], and (iii) affect how proteins are translated. Indeed, many bacterial and some eukaryotic viruses with large double-stranded DNA genomes commonly encode a range of tRNA genes[4,12–14]. While these genes are presumed to boost the translational efficiency and virulence in diverse hosts[15,16], their importance during infection remains to be demonstrated experimentally. Giant mimiviruses, whose genomes approach the size of small bacterial genomes, carry many typically cellular genes including those for translation initiation, elongation, and termination, as well as a range of aminoacyl-tRNA synthetases[17–19], among which methionine- and tyrosine-tRNA synthetases have been functionally characterized[20]. A single tRNA synthetase gene is also encoded in the genome of the largest known bacterial virus, *Bacillus megaterium* myovirus G[21], but its function lacks experimental confirmation. Some marine phages encode peptide deformylases, which are involved in post-translational modification[22] that, at least in cyanophages, may help preferentially produce the phage-encoded D1 photosystem protein[23]. Finally, T7-like podoviruses encode serine/threonine kinases that have been shown to phosphorylate around 90 proteins, including several involved in protein translation, such as host-encoded ribosomal proteins bS1 and bS6, translation initiation factors IF1, IF2, and IF3, and elongation factors G and P[24,25]. It was suggested that phosphorylation of these proteins may stimulate translation of the phage late mRNAs. Although it is now clear that viruses have evolved different strategies to tinker with protein translation, the genes encoding proteins that directly participate in the formation of the ribosomes are not yet observed in the genomes of cultured viral isolates. In fact, this feature—ribosome-encoding or not—has been proposed to signify a major divide between cellular life forms and viruses[26,27]. However, viral genome fragments assembled from environmental viral community sequence datasets (viral metagenomes), which vastly expand upon cultured sequence space, suggested that viruses might encode ribosomal proteins, specifically, bS1 and bS21. Though challenges insuring removal of contamination from cellular genomes and the lack of host context available warrants caution about such observations of "cellular features" in metagenome-only datasets[22,28], the findings are intriguing.

Here we leverage the greater genomic context now available from large-scale metagenomes and genomes to revisit the question of whether viral genomes encode ribosomal proteins (RPs). We identify 14 different RPs across viral genomes arising from cultivated viral isolates and metagenome-assembled viruses. We show that viruses tend to encode RPs known to be easily exchangeable between ribosomes, suggesting these proteins can replace cellular versions in host ribosomes, and confirm this experimentally for the two most common virus-encoded RPs, bS21 and bL12. Ecological distribution of virus-encoded RPs suggests certain level of ecosystem adaptations as aquatic viruses and viruses of animal-associated bacteria are enriched for different subsets of RPs. Overall, these results further blur the borders between viruses and cellular life forms.

## Results

**Ribosomal proteins encoded in cultivated virus genomes.** To systematically investigate the presence of RP-encoding genes in viral genomes, we first searched available reference genomes of cultivated viruses. Of 106 RP domains (Supplementary Table 1) that seeded our searches, 5 were identified across 16 viral genomes (Table 1). The genes were generally embedded within variable genomic contexts, even for homologous RP genes (Supplementary Fig. 1). Note that throughout this article we use the unified RP nomenclature[29], where capital letters "S" and "L", respectively, indicate whether the protein is present in the small or large ribosome subunit, whereas the lowercase letters denote that the protein is specific to bacteria (b), eukaryotes/archaea (e), or are universal (u).

We first identified a ribosomal protein eS30 domain, a component of the small 40S ribosomal subunit[30], in the Finkel–Biskis–Reilly murine sarcoma virus (FBR-MuSV), a member of the family *Retroviridae*. This domain was part of the *fau* gene fused to an N-terminal ubiquitin-like domain (Supplementary Fig. 2a). FBR-MuSV has acquired the cDNA copy of *fau* in inverse orientation, and production of the antisense RNA suppresses expression of endogenous *fau* mRNA, which leads to apoptosis inhibition and induces tumorigenesis[30,31]. Although the viral protein is not translated[30], the antisense transcript affects the production of the cellular *fau*[31] and thus might have an indirect effect on the ribosome biogenesis.

The remainder of the virus-encoded ribosomal proteins—bS21, bL9, bL12, and ribosome hibernation promotion factor (HPF)—were found in bacterial viruses (bacteriophages) infecting proteobacteria (from three different classes) and mycobacteria (phylum Actinobacteria) hosts (Table 1). Though RP-encoding bacteriophage genomes ranged in size between 44.5 and 358.6 kb, only large (>130 kb) genomes encoded bS21 and HPF, whereas only smaller (<80 kb) genomes encoded bL9 and bL12 (Table 1). However, many more phage genomes encoding RP would be needed to verify the significance of this observation.

The bS21 homolog was identified in pelagiphage HTVC008M, a myovirus. bS21 is a conserved component of the bacterial 30S ribosomal subunit (Fig. 1a) required for the initiation of polypeptide synthesis and mediates the base-pairing reaction between mRNA and 16S rRNA[32]. The viral protein was most similar (54% identity over the protein length) to the corresponding protein of its host, *Pelagibacter ubique* (Fig. 1b), an abundant member of the SAR11 clade (class Alphaproteobacteria), which is considered to represent one of the most numerous bacterial groups worldwide[33]. Maximum likelihood phylogenetic analysis showed that bS21 homologs from different families of alphaproteobacteria cluster together and form a sister group to the mitochondrial homolog, consistent with the scenario under which mitochondria have evolved from an alphaproteobacterial ancestor. In this tree, all alphaproteobacterial sequences are basal to the viral protein, strongly suggesting that the phage gene was horizontally acquired from the *Pelagibacter* host (Fig. 1c).

Ribosomal protein bL9 was identified in *Mycobacterium* phage 32HC, a siphovirus. bL9 binds to the 23S rRNA and is a component of the large 50S ribosome subunit (Supplementary Fig. 3a). The protein is involved in translation fidelity and is required to suppress frameshifting, and stop codon "hopping"[34]. bL9 has a highly conserved architecture consisting of two widely spaced globular RNA-binding domains connected by an elongated α-helix[35]. While the C-terminal domain in the viral bL9 homolog has been apparently non-homologously replaced with a

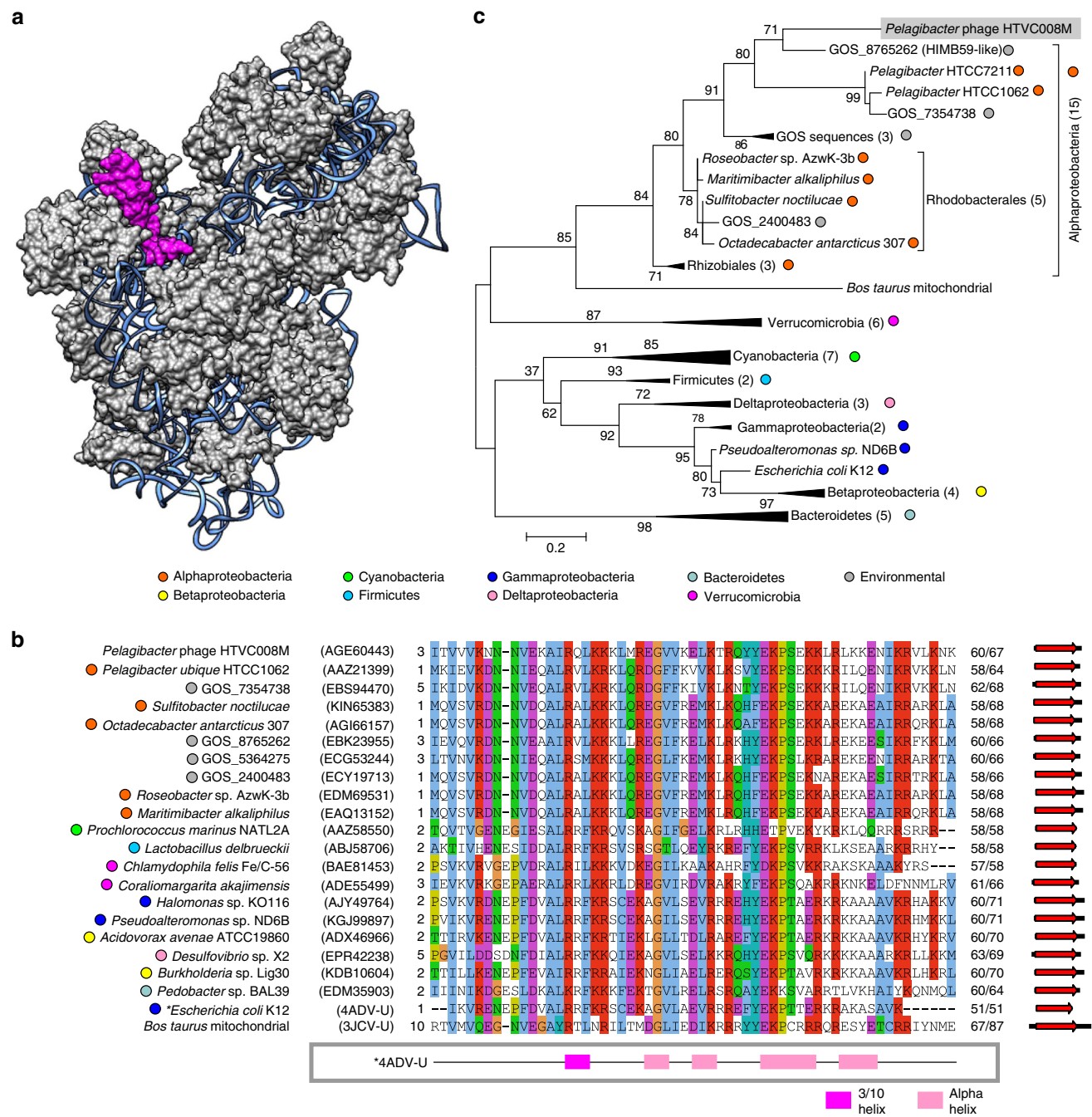

**Fig. 1** Virus-encoded ribosomal protein bS21. **a** Structure of the *Escherichia coli* 30S ribosomal subunit (PDB id: 4ADV). 16S ribosomal RNA is shown as blue ribbon. bS21 ribosomal protein is highlighted in pink. **b** Alignment of the ribosomal protein bS21 encoded by pelagiphage HTVC008M with homologs from representatives of distinct bacterial taxa and environmental sequences obtained from the Global Ocean Sampling (GOS) dataset. **c** Phylogenetic tree of ribosomal protein bS21. Taxonomic affiliations are represented by colored circles (see panel (**b**) legend)

sequence that lacks known function, the N-terminal RNA-binding domain and part of the α-helical spacer are preserved (Supplementary Fig. 2b), suggesting that the viral protein may bind to the 23S rRNA.

The next ribosomal protein encoded in sequenced viral genomes was bL12, which was found in 7 phages infecting proteobacteria from three different classes (Table 1). The bL12 proteins participate in the formation of the so-called bL12 stalk, a clearly defined morphological feature in the *E. coli* 50S ribosomal subunit, which besides bL12, contains ribosomal proteins uL10 and uL11 as well as the uL10- and uL11-binding region of the 23S rRNA[36] (Supplementary Fig. 3a). The phage-encoded bL12

domains are similar (~30–40% identity) to bona fide cellular ribosomal homologs and contain conserved residues involved in interaction with uL11 and elongation factors EF-G and EF-Tu (Supplementary Fig. 4a). Although in some phages (e.g., *Ralstonia* phage RSB3), the bL12 domain spans the entire protein, it was more common to observe these domains variably positioned within much larger polypeptides (up to 724 aa-long; Supplementary Fig. 4b). Notably, searches seeded with sequences flanking the bL12 domain in phage proteins resulted in identification of multiple phage homologs which specifically lack the bL12 domain (Supplementary Fig. 5). For example, proteins encoded by *Salmonella* phages FSL_SP-058 and FSL_SP-076 contain the

**Table 1 Ribosomal protein domains found in cultivated viruses**

| Domain | Protein | Name (family) | Genome length, kb | RP accession, length (aa) | Coverage, identity (%) | HHpred Probability (%) | E-value |
|---|---|---|---|---|---|---|---|
| Ribosomal_S30 | eS30 | Finkel–Biskis–Reilly murine sarcoma virus (R) | 3,811 | NP_598374, 133 | 85, 86 | 99.92 | 2.2E-26 |
| Ribosomal_S21 | bS21 | Pelagibacter phage HTVC008M (M) | 147,284 | AGE60443, 67 | 59, 46 | 99.81 | 2.7E-19 |
| Ribosomal_L9_N | bL9 | Mycobacterium phage 32HC (S) | 50,781 | AHJ86298, 86 | 33, 40 | 98.32 | 4.3E-07 |
| Ribosomal_L12 | bL12 | Dinoroseobacter phage DFL12phi1 (P) | 75,028 | AHX01035, 106 | 74, 32 | 99.9 | 1.0E-23 |
| | | Erwinia phage Ea9–2 (P) | 75,568 | AHI60108, 724 | 9, 32 | 96.87 | 6.3E-03 |
| | | Ralstonia phage RSB3 (P) | 44,578 | BAN92321, 98 | 59, 32 | 99.77 | 2.2E-18 |
| | | Roseophage DSS3P2 (P) | 74,611 | ACL81275, 107 | 62, 28 | 99.44 | 3.7E-13 |
| | | Salmonella phage FSL SP-058 (P) | 72,394 | AGF88397, 418 | 16, 34 | 96.05 | 1.8E-01 |
| | | Salmonella phage FSL SP-076 (P) | 72,098 | AGF88198, 418 | 15, 36 | 96.21 | 2.6E-02 |
| | | Sulfitobacter phage phiCB2047-B (P) | 74,485 | AGH07436, 126 | 25, 47 | 97.06 | 1.8E-03 |
| Ribosomal_S30AE | HPF | Cronobacter phage vB CsaM GAP32 (M) | 358,663 | AFC21633, 111 | 71, 34 | 99.96 | 4.2E-28 |
| | | Enterobacteria phage vB EcoM-FV3 (M) | 136,947 | AEZ65272, 105 | 74, 35 | 99.92 | 1.5E-23 |
| | | Escherichia coli bacteriophage rv5 (M) | 137,947 | ABI79209, 105 | 74, 33 | 99.96 | 1.3E-27 |
| | | Escherichia phage 2 JES-2013 (M) | 136,910 | AGM12525, 105 | 74, 32 | 99.96 | 3.0E-28 |
| | | Escherichia coli O157 typing phage 14 (M) | 131,952 | AKE47110, 105 | 74, 33 | 99.96 | 3.4E-28 |
| | | Escherichia phage vB EcoM FFH2 (M) | 139,020 | AEZ65272, 105 | 74, 35 | 99.93 | 7.9E-24 |

*R* Retroviridae, *M* Myoviridae, *S* Siphoviridae, *P* Podoviridae, *HPF* ribosome hibernation promotion factor

bL12 domains, whereas homologous protein from *Escherichia* phage Pollock lacks this domain, despite conservation of the upstream and downstream regions (Supplementary Fig. 5). Furthermore, in different phage genomes, bL12 proteins were encoded within widely different genomic contexts (Supplementary Fig. 1). These observations suggest that bL12 domain has been acquired by different phages on multiple, independent occasions, with some of these genes possibly being fixed in the phage genomes.

The last ribosomes-associated protein encoded in sequenced viral genomes was the ribosome hibernation promotion factor (HPF), or *Ribosomal_S30AE* (PF02482) domain-containing protein, which was encoded by 7 phages infecting *Cronobacter* and *E. coli* (six closely related phages with 92–97% average nucleotide identity) (Supplementary Fig. 6). HPF proteins are expressed during stasis and under unfavorable growth conditions; HPF binds ribosomes to stabilize 100S dimers that inhibit translation to enable cells to control translational activity without costly alteration of the ribosomal pool[37]. Multiple sequence alignment shows high conservation of the viral and cellular HPF homologs (Supplementary Fig. 7), suggesting that the gene transfer has occurred in a relatively recent past. In the HPF phylogeny, homologs from *E. coli* phages cluster amidst gammaproteobacterial sequences. By contrast, the more divergent protein encoded by *Cronobacter* phage clusters with sequences from members of the phylum Firmicutes, though this association is confounded by a potential long-branch-attraction artifact (Supplementary Fig. 8).

**Ribosomal proteins detected in viral metagenomes.** To place these findings of cultivated virus-encoded RPs into broader ecological context, we searched 424,225 viral contigs from two global metagenomic datasets[8,38] for putative RPs using the same 106 sequence profiles (see Methods). Overall, 13 putative ribosomal protein genes were identified across 1403 contigs (Fig. 2, Supplementary Table 2, Supplementary Fig. 3). By matching CRISPR spacers, comparing k-mer nucleotide frequencies and performing BLASTn homology searches against reference cellular sequences, hosts could be predicted for 74 (5.3%) of the uncultivated viruses encoding 7 different RPs (Supplementary Data 1). Despite this relatively low fraction of predicted hosts, RP-encoding phages are already associated with 8 bacterial phyla (Proteobacteria, Firmicutes, Verrucomicrobia, Bacteroidetes, Deinococcus-Thermus, Thermotogae, Cyanobacteria, and Actinobacteria).

The bS21, bL12, and HPF, which were found in cultivated phages, were also detected in uncultivated phages, with bS21 homologs dominating (88%) the pool of RPs detected (Fig. 2, Supplementary Table 2). While found in only one cultivated phage (see above), maximum likelihood phylogeny and genome context comparison using these metagenomic data suggested that multiple virus-host exchanges of bS21 protein-coding genes have occurred, likely across various bacterial phyla (Fig. 3). Notably, bS21-encoding viruses were almost exclusively from aquatic samples (90% of bS21s detected). Such repeated transfers and enrichment in aquatic samples suggest that virus-encoded bS21 proteins likely can provide a direct fitness benefit to aquatic bacteriophages. By contrast, bL12 and HPF were found across a broad range of samples (Fig. 2 and Supplementary Figs 8 and 9), suggesting that their repeated acquisition could be beneficial in multiple types of conditions and hosts.

Another 10 RPs detected in uncultivated viruses were not previously identified in isolate genomes (Supplementary Table 2). Commonly (>10 viral contigs) detected among these are bL31 and bL33. Although the biological function of bL33 remains

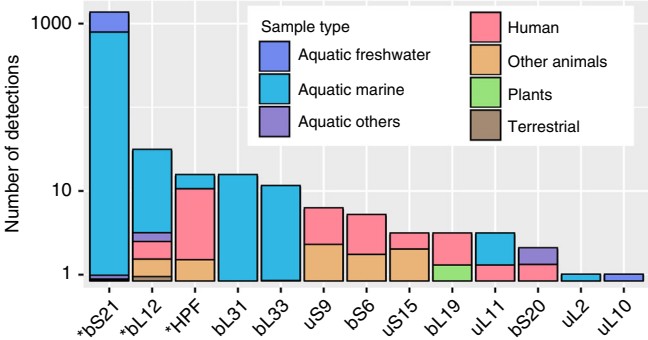

**Fig. 2** Detection of ribosomal proteins in uncultivated viral genomes. For each ribosomal protein detected, the total number of detection is shown on the y-axis (log$_{10}$ scale), and the bar is colored according to the type of samples in which this protein was detected (the sizes of the colored parts are proportional to the number of detections made in each type of samples). Ribosomal proteins also identified in cultivated viruses are identified with stars

obscure[39], it appears to contact tRNAs in the ribosomal E(exit)-site[40] (Supplementary Fig. 3b), whereas bL31, similar to HPF, plays a role in 100S formation, 70S association, and translation[41]. As in the case of bS21, viral contigs encoding bL31 or bL33 were almost exclusively detected in aquatic environments (Fig. 2). Maximum likelihood phylogenies and genome context comparisons highlighted a consistent pattern of at least two independent events of virus-host transfers involving viruses infecting different bacterial phyla (Supplementary Figs 10 and 11).

Thus, at this point, there is an emerging picture that ribosomal protein genes acquired through random sampling of host DNA might, in some cases, become fixed in viral genomes. Most (>99%) of the viruses contained only a single ribosomal protein gene (exception: 8 uncultivated viral contigs that contained 2; Supplementary Fig. 12), which is clearly not enough for viruses to build functional ribosomes on their own. Presumably, these viruses are merely tweaking ribosomal functioning in their hosts —just as observed for auxiliary metabolic genes (AMGs) whereby viruses typically do not encode complete pathways, but instead only select genes critical for the takeover and/or reprogramming of the host cell[6,7,42].

**RP-encoding genes are carried by temperate and lytic viruses.** Given that tRNA genes are more frequently encoded by lytic, as opposed to temperate, viruses[15], we sought to evaluate whether the acquisition and maintenance of RP-encoding genes is also linked to either the lytic or temperate life-style. Among the 16 RP-encoding cultivated viruses only FBR-MuSV and *Myco-bacterium* phage 32HC are temperate (see Methods). Specifically, FBR-MuSV, like all retroviruses, integrates into the cellular genome as an essential step during replication, whereas phage 32HC encodes a tyrosine recombinase (YP_009009518) predicted to integrate the viral genome into a host tRNA gene[43]. For the majority of RP genes encoded by uncultivated viruses there is not enough genomic data to ascertain their preferential association with either temperate or lytic viruses (73% are partial genomes <20 kb). Of 1396 uncultivated virus genome fragments, only 14 were detectably temperate, although these putative temperate viruses encoded 9 of the 13 virus-encoded RPs (Supplementary Table 3). Notably, the bS21 genes appear to be largely encoded by lytic viruses: of the 1310 genomes and large genomic contigs of cultivated and uncultivated viruses encoding bS21, only three (0.23%) genome fragments carry signature genes of temperate viruses (Supplementary Table 3). These results suggest that RP-

encoding genes can be acquired by both temperate and lytic viruses, with some of the genes possibly displaying preferential association with lytic viruses. However, additional genomic data will be required to quantitatively assess the dynamics of the RP gene flow between hosts, temperate viruses, and lytic viruses.

**Virus-encoded RP genes are under strong purifying selection.** Presence of ribosomal protein genes in viral genomes raises a question of what their functions in the course of the infection cycle might be and how do viruses benefit from carrying such genes. The eS30-encoding gene increases the transformation capacity of FBR-MuSV in vitro by twofold, providing clear fitness advantage to the virus[30]. It is conceivable that homologs of other ribosomal proteins might be also beneficial for the bacteriophages that encode them. For instance, it is known that bS21 is necessary during translation initiation step and in the absence of bS21, ribosomes are incapable of binding natural mRNAs[32]. Thus, phage-encoded bS21 might compete with and replace the cellular bS21, ensuring translation of viral transcripts. Similarly, viral bL12 domain proteins might provide interfaces for virus-specific translation factors. Protein bL9 is required for translational fidelity and is involved in suppression of frameshifting. In many members of *Caudovirales*, production of certain tail components is dependent on programmed translational frameshifting[44] and a viral copy of bL9 might help to achieve optimal frameshifting in these genes. Finally, it has been demonstrated that stalling of phage protein synthesis is one of the major defense strategies in Bacteroidetes[45]. Thus, viral homologs of HPF and bL31 might compete with the cellular homologs and prevent formation of ribosome dimers, thereby releasing translation inhibition and ensuring that phage transcripts are efficiently translated.

Given what seemed to be reasonable explanations for why viruses might benefit from encoding such genes, we next investigated whether virus-encoded RP genes appeared to be functional. To this end, we calculated the ratio of non-synonymous polymorphisms per non-synonymous site (pN) to the number of synonymous polymorphisms per synonymous site (pS). Here we used this metric to test if virus-encoded RPs were under purifying or positive selection, where the former (pN/pS <1) would indicate selection for a functional protein and the latter (pN/pS > 1) would indicate that the gene might be in the process of being phased out from the viral genome[46]. We found that well-sampled viral-encoded RP genes (>10× coverage, and ≥1 single nucleotide polymorphism, or SNP) had an average pN/pS = 0.10, with 84% having a pN/pS ≤ 0.20 (Supplementary Table 2). This suggests that these genes are under strong purifying selection and thus likely retained an important function after being transferred into virus genomes.

**Virus-encoded RPs are incorporated into ribosomes.** Although encouraging, the results of these in silico functional assays did not exclude the possibility that the viral RPs function in a different framework compared to their bona fide cellular homologs. Thus, we next explored whether the viral proteins are incorporated into ribosomes, by focusing on 3 RPs encoded by cultivated phages and most frequently detected in uncultivated phage genomes (Fig. 2). These were pelagiphage-encoded bS21, bL12 from *Salmonella* phage FSL SP-076, and HPF from *Escherichia coli* phage rv5. Following moderate and controlled expression of the respective viral proteins, 70S ribosomes were isolated under high-stringency salt conditions (see Methods) to avoid unspecific association of viral proteins[47]. Judging from the obtained ribosome profiles (Fig. 4a) and transmission electron microscopy (Supplementary Fig. 13), expression of the viral proteins did not affect the 70S stability. All examined

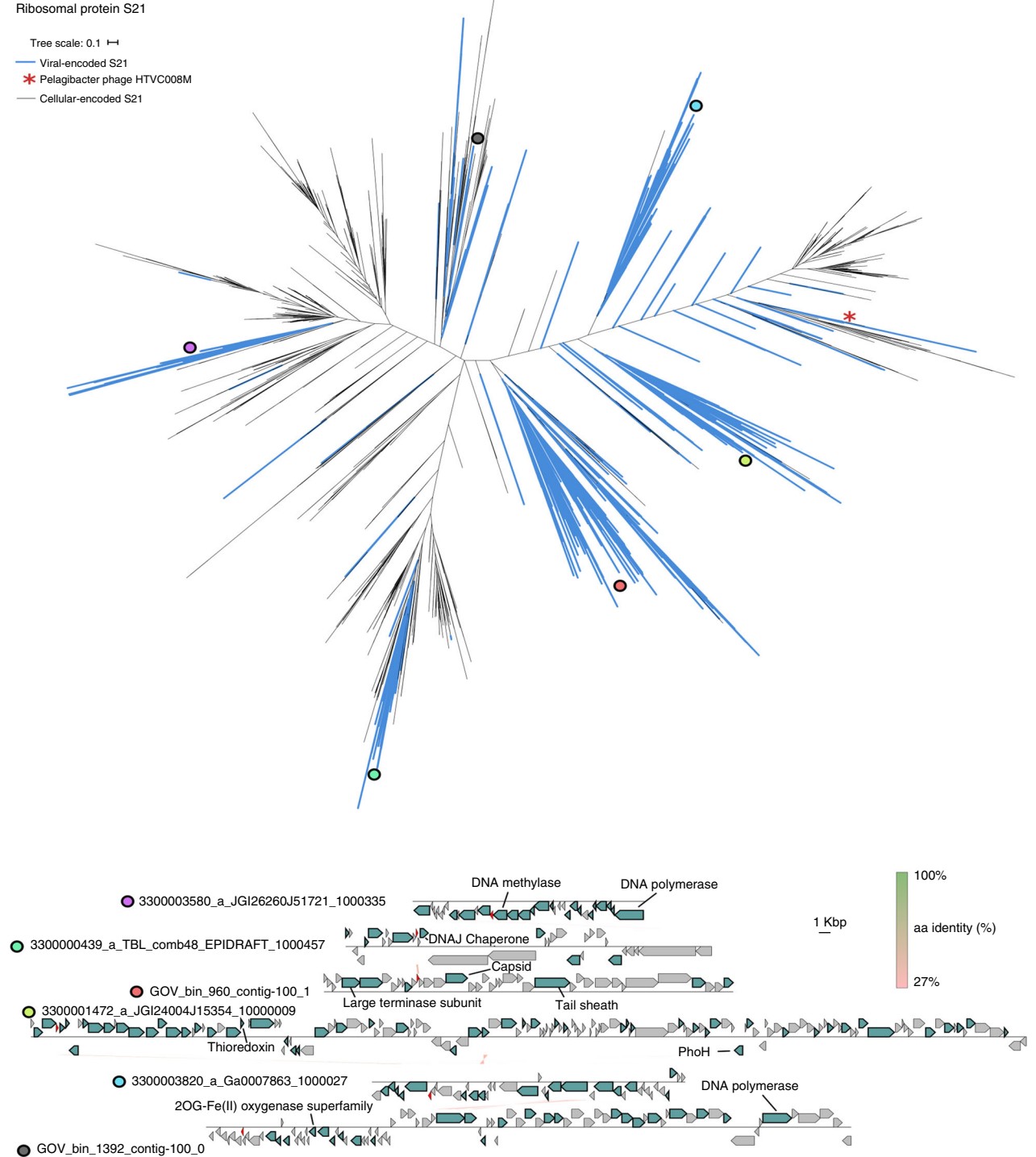

**Fig. 3** Ribosomal proteins bS21 identified in uncultivated viral genomes. Top: Phylogenetic tree of ribosomal protein bS21. Viral sequences are highlighted with blue branches. Bottom: Genome comparison of viral contigs encoding ribosomal protein bS21. Comparisons were done at the amino acid level, with the % identity displayed with a color scale. The predicted bS21-encoding genes are indicated in red. The position of these sequences in the tree (top panel) is indicated with colored circles

samples nearly exclusively contained 70S monoribosomes and a small portion of 100S particles (Fig. 4a). Subsequent mass spectrometry (MS) analysis of the 70S and 100S ribosomes purified on sucrose gradients unequivocally showed that bS21 and bL12 (Table 2, Supplementary Data 2 and 3), but not HPF (Supplementary Data 4), were stably incorporated into the ribosomes when expressed in *E. coli*. Consistently, there was no discernible effect on the formation of 100S particles in cells

expressing the viral HPF homolog (Fig. 4a). Notably, HPF was detected using MS in the crude cell extracts (Table 2, Supplementary Data 5), indicating that lack of its incorporation into ribosomes is not due to poor protein expression, but may rather result from other factors, such as inadequate growth phase, genuine loss of ability to bind to ribosomes or dissociation due to stringent washes with salt during ribosome isolation.

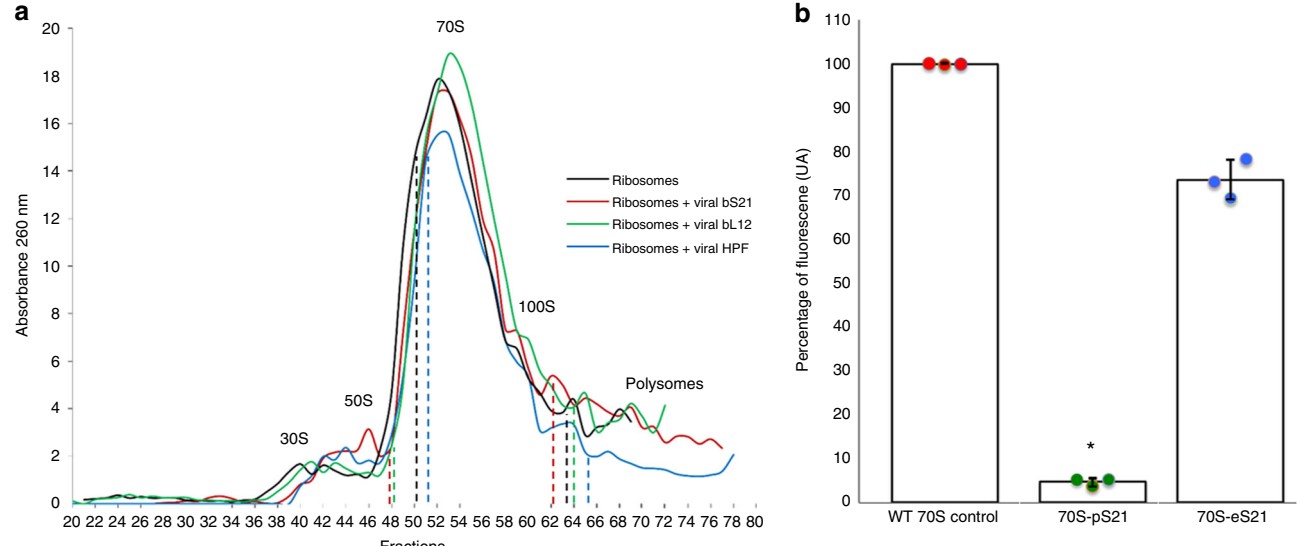

**Fig. 4** Ribosome analysis of extracts from NM522 *Escherichia coli* cells. **a** Sedimentation profiles of NM522 *E. coli* ribosomes. Wild-type NM522 *E. coli* cells (black curve) and cells expressing viral bS21 (red), bL12 (green) or HPF (blue) were lysed and their ribosomes were purified using a 10–50% sucrose gradient (see experimental section). The dotted lines indicate the fractions that were pooled and further analyzed by mass spectrometry. **b** Quantification of in vitro translation of GFP by *E. coli* 70S ribosomes carrying either *E. coli* wt bS21 (control), *E. coli* streptavidin-tagged bS21 (70S-eS21) or viral streptavidin-tagged bS21 (70S-pS21). Translation assay was performed using PURExpress® ΔRibosome Kit, complemented with 10 pmol of purified ribosomes and 250 ng of a PCR product encoding for GFP under control of T7 promoter. Fluorescence signal was detected by spectrofluorimetry at 510 nm with an excitation at 485 nm. The percentage of fluorescence was measured with respect to the translation control. The error bar represents the standard deviation measured over three independent experiments. The variance was analyzed using Kruskal–Wallis test followed by a Dunn's multiple comparison test (*p*-value for 70S-pS21 = 0.0219)

---

**Table 2 bS21, bL12, and HPF peptides identified by mass spectrometry in ribosome preparations and crude cell extract**

| Protein, source | Accession number | Peptides in ribosomal fractions | Peptides in cell extract |
|---|---|---|---|
| bS21, Pelagibacter phage HTVC008M | AGE60443 | SITVVVKNNNVE, KKLRLKKE | N.D. |
| HPF, Escherichia coli phage rv5 | ABI79209 | None | GSDAYEATDR, VENDHQEVMAFIFDNSGK, VENDHQEVM*AFIFDNSGK, VKIDFGE |
| bL12, Salmonella phage FSL SP-076 | AGF88397 | VNDDTETYYIDLPYVAR | N.D. |

*oxidized form of methionine
*N.D.* not determined

---

In order to check whether the incorporation of virus-encoded ribosomal proteins alters bacterial translation, we next proceeded to purification of ribosomes carrying bS21 and bL12 viral proteins. Towards this aim, we overproduced the two streptavidin-tagged viral ribosomal proteins in *E. coli* and the ribosomes containing the tagged viral proteins were purified by affinity. Only ribosomes containing bS21 could be purified, suggesting that the tagged bL12 cannot be efficiently incorporated or that its interaction with the ribosome is not sufficiently strong to allow the purification. The presence of the viral bS21 within the ribosomes was confirmed by MS analysis. The activity of the purified ribosomes was then verified by an in vitro translation assay. The protein synthesis is severely impaired in the presence of viral bS21 (~5% compared to wild-type ribosomes; Fig. 4b). This impairment is not due to the presence of the affinity tag, because ribosomes containing the streptavidin-tagged *E. coli* bS21 were still active (~75% compared to wild-type ribosomes; Fig. 4b). The inactivity of ribosomes carrying the viral bS21 suggests that additional virus- or host-encoded factors may be needed for proper translation or that the RP from a phage infecting

*Pelagibacter ubique* may not be fully compatible with *E. coli* ribosomes.

Regardless, these results indicate that following their transfer into viral genomes, bS21 and bL12 retained the ability to incorporate into ribosomes, successfully competing with the native cellular counterparts. Whether incorporation of these and other viral RPs modulates protein translation during phage infection remains to be demonstrated by further functional assays both with heterologously expressed RPs as well as in the framework of the infection with the corresponding phages.

## Discussion

The evolution of life is a history of virus-host interactions, an incessant "arms race" between viruses and cells[48,49]. To ensure their successful proliferation, both parties have evolved numerous molecular strategies which are continuously being uncovered. Among the most remarkable recent discoveries are various defense and counter-defense mechanisms[9], peptide-based communication strategies employed by bacterial viruses to alternate

between the lytic and lysogenic life cycles[50], as well as virus-mediated manipulation of the behavior and reproductive strategies of the hosts[51,52]. Horizontal gene transfer plays a key role in these processes, with many, if not all, molecular innovations being shuttled back and forth between viruses and cells, irrespective of who the original inventor was[53]. Indeed, even such hallmark virus components as the major virion proteins have been recurrently exapted from cellular proteomes and occasionally reintroduced to perform diverse cellular functions[54]. Similarly, numerous AMGs have been recruited by viruses from their hosts, enabling viral manipulation of various cellular pathways, and transforming the infected cell into a specialized virion factory. Genes for the core components of translation machinery, namely RPs, until now appeared as the last unbreached boundary between the cellular and viral kingdoms, despite the fact that certain viruses were known to tinker with protein translation by encoding tRNA genes, protein deformylases or tRNA synthetases. Our current work shows that diverse ribosomal proteins are in fact commonly encoded by numerous cultivated and uncultivated viruses with relatively small genomes and offers support for them having an evolutionary fitness advantage for viruses during infection. Notably, virus-encoded RPs appear to be differentially selected for across environments as aquatic viruses are enriched for bS21, bL31, and bL33, whereas phages of animal-associated bacteria are enriched for bS6, uS9, uS15, and HPF (Fig. 2). Although ribosomes are highly stable macromolecular assemblies which retain most of their original components during cellular growth and division[55], some elements (proteins bS21, bL12, bL9, bL31, and bL33) are highly dynamic, solvent accessible, and among the few proteins that are loosely bound to the ribosome and can be exchanged in vivo between ribosomes[55,56]. These dynamic ribosomal proteins are enriched in viruses, presumably because they are most suited to homologous replacement during infection. Just as in ocean virus AMGs, it appears that viruses co-opt and fix not all enzymes in a metabolic pathway, but instead only central regulators or enzymes for key rate-limiting steps in a pathway[6,8,57]. More generally, such selective acquisition of key components of the multisubunit assemblies, such as ribosomes, or recruitment of central regulators of rate-limiting steps in metabolic pathways appears to be a general strategy employed by viruses to optimize the metabolic state of the infected cells and/or to achieve the takeover of the host. Together, these functional and computational findings highlight widespread means by which viruses might modulate translation upon infection and either call into question a commonly used definition for life—the presence/absence of RPs—or further blur the borders between viruses and cellular life forms. Given that viral metagenomics becomes a major approach in virus discovery[58] with new RNA and DNA virus genomes and large genome fragments being discovered by the thousands to tens of thousands[8,38,59,60], we are most certainly bound to discover many more new strategies of host take-over in the near future.

## Methods

**Sequence analyses**. All viral genomes were downloaded from viral RefSeq database (ftp://ftp.ncbi.nlm.nih.gov/refseq/release/viral/). A hidden Markov model (HMM) profile was downloaded from the PFAM database (http://pfam.xfam.org/) for each domain listed in Supplementary Table 1. In total, 106 sequence profiles corresponding to distinct ribosomal protein domains were used as seeds to search the proteomes of viruses infecting hosts from the three cellular domains, as well as proteins predicted on viral contigs from two previously published global metagenomic datasets, Global Ocean Virome[8], and Earth's Virome[38], which are available at https://img.jgi.doe.gov/cgi-bin/vr/main.cgi and http://datacommons.cyverse.org/browse/iplant/home/shared/iVirus/GOV. Notably, domain S1, which is repeated 4 to 6 times in the ribosomal protein bS1, is not exclusive to RPs as it is common across diverse RNA-binding proteins and fused to non-ribosomal functional motifs (pfam id: PF00575.18). Thus while domain S1 was found in homologs of vaccinia virus interferon inhibitor K3L[61], which is conserved in chordopoxviruses belonging

to 7 different genera, it was not considered further due to potential functional ambiguity. The domains were identified by HHsearch[62] with E-value of 1e-5. For isolates, the identified hits were then manually inspected using HHPRED[62]. All alignments were constructed using PROMALS3D[63]. Maximum likelihood phylogenetic trees were constructed using PhyML[64] using a WAG substitution model and the proportion of invariable sites estimated from the data. For metagenomic predicted proteins, multiple alignments were built with Muscle[65] and maximum likelihood phylogenetic trees were computed with FastTree[66], and displayed with iToL[67]. Genomic comparisons were performed using BLAST with the BLOSUM45 matrix. The ribosomal structure was downloaded from PDB database and visualized using Chimera[68].

Putative temperate phages were identified by searching for the following PFAM domains in ribosomal protein-encoding viral contigs: Mu-transpos_C, Phage_int_SAM_5, and Phage_integrase (hmmsearch, threshold of 30 on bit score). Host predictions for viral metagenomic contig were obtained from the original studies (i.e., ref. [8] and [38]) and, in the case of the Earth's virome[38], complemented with a k-mer based prediction computed with WIsH[69] (p-value threshold of 0.001).

To further confirm the functionality of RPs encoded on uncultivated viral genomes, selective constraint on these AMGs was evaluated through pN/pS calculation, as in ref. [46]. Briefly, synonymous and non-synonymous SNPs were observed in each ribosomal protein gene covered ≥10 × , and compared to expected ratio of synonymous and non-synonymous SNPs under a neutral evolution model if at least 1 SNP was identified. The interpretation of pN/pS is similar as for dN/dS analyses, with the operation of purifying selection leading to pN/pS values <1.

**Genetic constructions**. The genes encoding for bS21 protein from *Pelagibacter* phage HTVC008M (AGE60443), HPF protein from *Escherichia coli* bacteriophage rv5 and bL12 protein from *Salmonella* phage FSL SP-076 (AGF88397) were synthetized by Eurofins Genomics (Ebersberg, Germany). bS21 and HPF genes were cloned into pEX-A2 plasmid and bL12 gene into pEX-K4 plasmid. The gene corresponding to HPF viral protein was digested by BsaI and HindIII and inserted into a pBAD24 vector between NcoI and HindIII restriction sites. The genes corresponding to bS21 and bL12 viral proteins were cloned into the same vector, using EcoRI and HindIII restriction sites. The C-terminal streptavidin-tagged versions of bS21 genes from *Pelagibacter* phage and *E. coli* were also synthetized and cloned into pBAD24 plasmids. The pBAD24 plasmid harbors an arabinose dependent promoter, a pBR322 origin and the ampicillin resistance coding sequence.

**Protein expression and cell retrieval**. *Escherichia coli* strain NM522 was used for expression of viral bS21, HPF, and bL12 proteins. The same strain harboring empty pBAD24 was used as a negative control. Overnight pre-cultures were grown in the presence of 1 mM of L-arabinose and 100 µg/mL of ampicillin. Then the expression was maintained in the cell culture until the end of exponential phase. Once the cultures reached an $OD_{600nm}$ of 1, the cells were centrifuged at 8700× g for 7 minutes at 4 °C. The cell pellet was then washed into saline water at a concentration of 9 g/L of NaCl. A second centrifugation was made and the bacterial pellet was frozen at −80 °C.

**70S Ribosome purification**. The *E. coli* cells were resuspended in Buffer 1 (Tris-HCl pH7,5 20 mM, MgOAc 50 mM, NH4Cl 100 mM, EDTA 0.5 mM and DTT 1 mM) and finally lysed using the French Press. The lysate was centrifuged and the supernatant was put above the same volume of high-salt sucrose buffer (Tris-HCl pH7.5 10 mM, MgCl2 10 mM, NH4Cl 500 mM, EDTA 0.5 mM, certified RNase free sucrose 1.1 M and DTT 1 mM) in order to wash the ribosomes. After centrifugation at 93,000× g for 20 h at 4 °C using Type 70 Ti rotor (BECKMAN L-90 ultracentrifuge), the ribosomes form a translucent pellet. The ribosome pellet was washed several times to remove membranes and then resuspended in Buffer 2 (Tris-HCl pH7.5 10 mM, MgCl2 10 mM, NH4Cl 50 mM, EDTA 0.5 mM and DTT 1 mM) on ice. An equivalent of $200OD_{260nm}$ units of ribosomes were loaded on top of a 10–50% sucrose gradient into polycarbonate tubes. The ultra-centrifugation was performed at 95,000× g, for 18 h at 4 °C using SW28 rotor (BECKMAN L-90 ultracentrifuge). The gradient was then fractionated into 500 µL aliquots. The $OD_{260nm}$ values were determined for each fraction to locate the 70S absorbance peak. The corresponding fractions were pooled in one volume of buffer 2 and centrifuged at 93,000× g for 20 h at 4 °C using Type 70 Ti rotor in order to remove sucrose. The pellet was recovered in buffer 2 and after titration, the ribosomes were ready for mass spectrometry analysis or purification using StrepTrap™ HP columns (GE healthcare) for in vitro translation assays.

The StrepTactin Sepharose column (StrepTrap™ HP columns, GE Healthcare Life Sciences) was equilibrated with the buffer 3 (MgOAc 9 mM, NH4Cl 10 mM, KCl 50 mM, HEPES-KOH pH7, 5 5 mM and DTT 1 mM). After injection of the purified ribosomes, the column was washed with 100 mL of buffer 3 before elution with 2.5 mM of d-Desthiobiotin. Fractions containing ribosomes were finally concentrated (Amicon 30 kDa) and resuspended in the buffer 3 for in vitro translation assays.

**In vitro translation assays.** In vitro translation was performed using PUREx-press® ΔRibosome Kit (New England Biolabs). Wild-type ribosomes (controls) or eluted ribosomes carrying *E. coli* or virus-encoded tagged bS21 were tested (final concentration: 10 pmol). A green fluorescent protein was translated by using a PCR product containing T7 promoter to rapidly evaluate the translation rates.

**Fluorescence analysis.** After 3 h incubation at 37 °C, the reaction volume was adjusted to 125 µL and distributed in cuvettes for Fluorescence measurement with LS 55 Fluorescence Spectrometer (PerkinElmer). Fluorescence intensity of translated GFP was determined using the FinLab software with following conditions: λ excitation 485 nm/slit 10/photomultiplicator 775/λ emission 510 nm.

**Negative staining.** Following ribosome separation, we diluted samples 10 times in Buffer 2 and applied them to freshly glow-discharged 300-mesh collodion/carbon-coated grids. After three washes in this buffer, grids were stained with 2% uranyl acetate for 30S. The grids were then observed with a Tecnai G2 Sphera transmission electron microscope operating at 200 kV. Images were recorded with a 4000 × 4000 Gatan Ultrascan 4000 CCD camera at a nominal magnification of ×50,000.

**Liquid digestion of ribosomal samples.** 25 µg of ribosomes were digested according to the following protocol: first, 53.5 µl of 50 mM ammonium bicarbonate buffer (pH 7,8) was added to the sample to 65 µL total volume. After vortexing 1 minute, tubes were incubated 10 minutes at 80 °C and then sonicated for two minutes. Reduction of disulfide bonds step was processed by adding 12.5 µl of 65 mM DTT to the sample and was incubated 15 minutes at 37 °C after agitation 1 minute. Alkylation of reduced disulfide bonds was realized by adding 135 mM iodoacetamide. Microtube was then incubated 15 minutes in the dark at room temperature, under agitation. Finally, proteins were digested overnight at 37 °C with 10 µl of either modified endoproteinase glu-c ([0.1 µg/µl.], Promega, Madison, WI) in 50 mM ammonium bicarbonate buffer for bS21 (due to high lysine and arginine content in bS21) or with modified Trypsine ([0.1 µg/µl.], Promega, Madison, WI) in 50 mM ammonium bicarbonate buffer for HPF, bL12 and control.

**Protein prefractionation and digestion.** Twenty five micrograms of soluble crude protein extracts of *E. coli* were boiled for 10 min with 5 µl of LDS Sample buffer 4X and 2 µl of reducing agent (DTT 10 × (500 mM)). They were then separated on a NuPAGE® Novex® 4–12 % gradient Bis-Tris gel (Invitrogen Corparation, USA) in MES SDS Running Buffer (Invitrogen: 50 mM MES, 50 mM Tris-HCl, 1 % SDS, 1.025 Mm EDTA) using Xcell SureLock Mini Cell (Invitrogen).

Gel was stained with EZBlue (Sigma-Aldrich) for 30 min and destained with water overnight. Each gel lane was manually cut into 2 slices of approximately the same size in the region of 7 kDa–14 kDa. The slices were first treated with 50 mM NH4HCO3 in acetonitrile/water 1:1 (v/v), dehydrated with 100% acetonitrile and rehydrated in 100 mM NH4HCO3. Next they were washed again with 50 mM NH4HCO3 in acetonitrile/water, 1:1 (v/v) and dehydrated with 100% acetonitrile. The slices were then treated with 65 mM DTT for 15 min at 37 °C, and with 135 mM iodoacetamide in the dark at room temperature. Finally, the samples were washed with 100 mM NH4HCO3 in acetonitrile/water, 1:1 (v/v), and dehydrated with 100% acetonitrile before being rehydrated in 100 mM NH4HCO3, washed with 100 mM NH4HCO3 in acetonitrile/water, 1:1 (v/v) and then dehydrated again with 100% acetonitrile. Proteins were digested overnight at 37 °C with 4 ng/l of modified trypsin (Promega, Madison, WI) in 50 mM NH4HCO3. Peptides were extracted by incubating the slices first in 80 µl of acetonitrile/ water/trifluoroacetic acid (70/30/0.1; v/v/v) for 20 min, and then in 40 µl of 100% acetonitrile for 5 min and finally in 40 µl of acetonitrile/water/trifluoroacetic acid (70/30/0.1; v/v/v) for 15 min. Supernatants were transferred into fresh tubes and concentrated in a SpeedVac (Thermo Scientific) for 15 min to a final volume of 40 µl.

**LC-MS/MS analysis.** Shotgun analyses were conducted on a LTQ-Orbitrap XL (ThermoFisher Scientific) mass spectrometer. The MS measurements were done with a nanoflow highperformance liquid chromatography (HPLC) system (Dionex, LC Packings Ultimate 3000) connected to a hybrid LTQ-Orbitrap XL (Thermo Fisher Scientific) equipped with a nanoelectrospray ion source (New Objective). The HPLC system consisted of a solvent degasser nanoflow pump, a thermostated column oven kept at 30 °C, and a thermostated autosampler kept at 8 °C to reduce sample evaporation. Mobile A (99.9% Milli-Q water and 0.1% formic acid (v:v)) and B (99.9% acetonitrile and 0.1% formic acid (v:v)) phases for HPLC were delivered by the Ultimate 3000 nanoflow LC system (Dionex, LC Packings). An aliquot of 10 µL of prepared peptide mixture was loaded onto a trapping pre-column (5 mm × 300 µm i.d., 300 Å pore size, Pepmap C18, 5 µm) over 3 min in 2% buffer B at a flow rate of 25 µL/min. This step was followed by reverse-phase separations at a flow rate of 0.250 µL/min using an analytical column (15 cm × 300 µm i.d., 300 Å pore size, Pepmap C18, 5 µm, Dionex, LC Packings). We ran a gradient from 2−35% buffer B for the first 60 min, 35−60% buffer B from minutes 60−85, and 60−90% buffer B from minutes 85−105. Finally, the column was washed with 90% buffer B for 16 min and with 2% buffer B for 19 min before the next sample was loaded. The peptides were detected by directly eluting them from the HPLC column into the electrospray ion source of the mass spectrometer. An

electrospray ionization (ESI) voltage of 1.6 kV was applied to the HPLC buffer using the liquid junction provided by the nanoelectrospray ion source, and the ion transfer tube temperature was set to 200 °C. The LTQ-Orbitrap XL instrument was operated in its data-dependent mode by automatically switching between full survey scan MS and consecutive MS/MS acquisitions. Survey full scan MS spectra (mass range 400−2000) were acquired in the Orbitrap section of the instrument with a resolution of $r = 60\,000$ at m/z 400; ion injection times were calculated for each spectrum to allow for accumulation of $10^6$ ions in the Orbitrap. The ten most intense peptide ions in each survey scan with an intensity above 2000 counts (to avoid triggering fragmentation too early during the peptide elution profile) and a charge state ≥2 were sequentially isolated at a target value of 10,000 and fragmented in the linear ion trap by collision-induced dissociation. Normalized collision energy was set to 35% with an activation time of 30 ms. Peaks selected for fragmentation were automatically put on a dynamic exclusion list for 30S with a mass tolerance of ± 10 ppm to avoid selecting the same ion for fragmentation more than once. The following parameters were used: the repeat count was set to 1, the exclusion list size limit was 500, singly charged precursors were rejected, and the maximum injection time was set at 500 and 300 ms for full MS and MS/MS scan events, respectively. For an optimal duty cycle, the fragment ion spectra were recorded in the LTQ mass spectrometer in parallel with the Orbitrap full scan detection.

For Orbitrap measurements, an external calibration was used before each injection series ensuring an overall error mass accuracy below 5 ppm for the detected peptides. MS data were saved in RAW file format (Thermo Fisher Scientific) using XCalibur 2.0.7 with tune 2.4. The data analysis was performed with Proline software 1.4 supported by Mascot Distiller and Mascot server (v2.5.1; http://www.matrixscience.com) database search engine for peptide and protein identification using its automatic decoy database search to calculate a false discovery rate (FDR) of 1% at the peptide level. MS/MS spectra were compared to the *Escherichia coli* Reference proteome set database containing the phage ribosomal proteins (UniProt release 2017_01, 18 January 2017, 23022 sequences, 7070297 residues). Mass tolerance for MS and MS/MS was set at 10 ppm and 0.5 Da, respectively. The enzyme selectivity was set to full trypsin with one miscleavage allowed for samples HPF and bL12 and the enzyme selectivity was set to full V8-DE with one miscleavage allowed for sample bS21.

Protein modifications were fixed carbamidomethylation of cysteines, variable oxidation of methionine, variable acetylation of lysine, and variable acetylation of N-terminal residues.

**Reporting summary.** Further information on experimental design is available in the Nature Research Reporting Summary linked to this article.

## Data availability

The authors declare that the data supporting the findings of this study are available within the paper and its supplementary information files. Viral contigs analyzed in this study are from two previously published global metagenomic datasets (Global Ocean Virome and Earth's Virome) that are available at https://img.jgi.doe.gov/cgi-bin/vr/main.cgi and http://datacommons.cyverse.org/browse/iplant/home/shared/iVirus/GOV.

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

## Acknowledgements

This work was supported by grant ERC UE 340440 to PF; Agence Nationale pour la Recherche grants to M.K. (#ANR-17-CE15–0005–01) and R.G. (Direction Générale de

l'Armement; #ANR-14-ASTR-0001); the Virus-X project (EU Horizon 2020, No. 685778) to M.K.. C.M.M. was supported by the European Molecular Biology Organization (ALTF 1562-2015) and Marie Curie Actions program from the European Commission (LTFCO-FUND2013, GA-2013-609409); C.G. was supported by Direction Générale de l'Armement and Ministère de l'Enseignement supérieur et de la Recherche; M.B.S. was supported by Gordon and Betty Moore Foundation (#3305, 3790) and National Science Foundation (OCE#1536989) awards. F.R.-V. was supported by grant VIREVO CGL2016-76273-P [AEI/FEDER, EU] (cofounded with FEDER funds). The work conducted by the U.S. Department of Energy Joint Genome Institute, a DOE Office of Science User Facility, is supported under Contract No. DE-AC02-05CH11231. We thank Fanny Demay for the technical assistance and Sophie Chat for help with electron microscopy.

## Author contributions

M.K. conceived the study. C.M.M. and S.R. performed sequence analyses. C.M.M., S.R., M.B.S. and M.K. interpreted the results. C.G. and R.G. planned and performed functional assays. R.L. performed LC-MS/MS analysis. F.R.-V. and P.F. contributed resources. C.M.M. and M.K. wrote the first draft of the manuscript, with substantial contribution from S.R., M.B.S. and R.G. All authors edited and approved the final version of the manuscript.

## Additional information

**Competing Interests:** The authors declare no competing interests.

