## [Peer Review File · Nature Communications]

Reviewers' Comments:

Reviewer #1:

Remarks to the Author:

General Comments

The manuscript, 'Numerous cultivated and uncultivated viruses encode ribosomal proteins' by Mizuno et al. addresses the observation that phages and viruses can harbor ribosomal protein genes. The work is mainly based on bioinformatic analysis of phage and virus metagenomes and one sucrose density gradient coupled with mass spec to validate incorporation of ribosomal proteins encoded by the phage into the cellular protein synthesis machinery. Considering that there is constant exchange of genetic information between the host and the virus (as also discussed by the authors), this observation is in general not surprising. However, the fact might hint towards ribosome heterogeneity during virus infection that could result in selective translation of viral transcripts. This was already shown for metabolic genes taken up by viruses, which enable the formation of a specialized host cell programmed to stimulate virus particle production. With respect to protein synthesis some studies published even more than 20 years ago provide evidence for the regulation of host protein synthesis by phosphorylation of ribosomal proteins and initiation factors (these references are missing!). Thus, the idea proposed here lacks novelty. For publication in my opinion these hypotheses still require experimental support showing e.g. the alteration of the specificity of protein synthesis triggered by expression of viral encoded ribosomal proteins.

Specific points

Discussion: As mentioned above I miss several publications that provide evidence for the regulation of host protein synthesis by phosphorylation of ribosomal proteins and initiation factors (e.g. Robertson and Nicholson, Biochem. 1992)

Figure 4: Considering the high Mg²⁺ concentration of 10 mM, I am wondering why no peaks representing the ribosomal subunits are visible in the sucrose density gradient! Moreover, the pictures showing the negative staining of the 70S ribosomes do not add any information and could be omitted. In addition, the authors discuss the fact that the HPF is not present in the 70S fraction despite it is expressed and detectable in the cell extract. However, HPF is stabilizing 100S dimers, as the authors correctly mention in the manuscript on page 5, 2nd line! Thus, I would not expect the protein to be present in the 70S fraction!

Reviewer #2:

Remarks to the Author:

The authors present a study about ribosomal proteins (RPs) in cultivated and uncultivated viruses. As the authors state certain RPs have been associated with environmental viral sequences previously, however this study went deeper to ascertain their presence in a broader sense. They also show that at least 2 of the 14 proteins appeared to incorporate into the final protein structure when expressed in *E. coli*. However it is important that function was not shown in this study.

Line 52: 'take over' having one or two proteins of a complex large system is somewhat misleading, perhaps it is a modulation that is needed and/or there are other unknown imperative function(s) involved in virus/host associations during infection.

Lines 80 -81: Hyphen's as punctuation in a manuscript?

Lines 107-108: For the genome size:RP type correlation, can the others discuss this further. Is there any reason why this might be occurring? Was the cutoff arbitrary?

Lines 116-118: "...phylogenetic analysis strongly suggests that the phage gene was horizontally acquired ..." I would like the authors to discuss this more and/or provide more data to support the claim. Based on Figure 1c it can definitely be said that the host and phage reside in the same clade with good support, however it is unclear if the gene went from phage to host or the reverse (the latter suggested by the authors).

Lines 125-126: Does this suggest that the protein does not function and was merely an acquisition of genetic material during packaging?

Line 161: references Figure S3B. Are the authors correct to reference it here? It is unclear, the legend indicates the arrows point/indicate which RPs were found within viruses but does not specifically state environmental. There is also only 12, but the text says 14 were identified?

Lines 168-172: I don't see how Figure 3 truly supports the authors claim that there have been "7 distinct virus-host exchanges of bs21". Please provide more evidence or clarification.

Line 80: Is the 10 contigs coming from the pool of 78 that had a putative host assigned? Or the total 1,403?

Lines 188-189: "Thus, at this point, there is an emerging picture that viruses might randomly sample host DNA, including ribosomal protein genes, and that in some cases these might become fixed in viral genomes." This is not a new idea, but the way written it suggests it.

Lines 202-205: The sentence is repeated.

Lines 221-222: "Thus, phage-encoded bs21 might compete with and replace the cellular bs21, forcing preferential translation of viral transcripts." I feel like this is a big jump. There is no indication, in the presented data, of preference. However, it does suggest that it gives the virus its own mechanism for translation initiation without relying on host machinery (at least for this one step).

Lines 231-239: I really do not like the use of pN/pS used for determining or suggesting function. It is used for evolution, yes, but I don't see the direct link to function. Can the authors clarify why they chose this and also provide any necessary references that show a direct link between this inference and function.

Lines 278-280: "Nevertheless, genes for the core components of translation machinery, namely RPs, until now appeared as the last unbreached boundary between the cellular and viral kingdoms". This is confusing since it was stated in the introduction (Line 80-81) that "viral genome fragments ... suggested that viruses might encode ribosomal proteins. Therefore this is not new information for bs21 but it is definitely more indepth analysis.

Reviewer #1 (Remarks to the Author):

General Comments

The manuscript, 'Numerous cultivated and uncultivated viruses encode ribosomal proteins' by Mizuno et al. addresses the observation that phages and viruses can harbor ribosomal protein genes. The work is mainly based on bioinformatic analysis of phage and virus metagenomes and one sucrose density gradient coupled with mass spec to validate incorporation of ribosomal proteins encoded by the phage into the cellular protein synthesis machinery. Considering that there is constant exchange of genetic information between the host and the virus (as also discussed by the authors), this observation is in general not surprising. However, the fact might hint towards ribosome heterogeneity during virus infection that could result in selective translation of viral transcripts. This was already shown for metabolic genes taken up by viruses, which enable the formation of a specialized host cell programmed to stimulate virus particle production. With respect to protein synthesis some studies published even more than 20 years ago provide evidence for the regulation of host protein synthesis by phosphorylation of ribosomal proteins and initiation factors (these references are missing!). Thus, the idea proposed here lacks novelty.

RESPONSE: We thank the reviewer for the comment and suggestions. In the revised manuscript we included the information and relevant references on regulation of translation by phosphorylation. We also revised our manuscript to better highlight the novelty of our results. First, although there is indeed constant exchange of genetic information between hosts and viruses, very few of these genes are retained durably in viral genomes. The very fact that genes encoding ribosomal proteins in viral genomes are among this small group yet remained largely unnoticed until now, despite the availability of thousands of complete viral genomes in databases, is surprising. Second, the absence of ribosomal RNA and ribosomal protein genes in viral genomes has prompted scientists to consider these features when defining viruses, their origins and relationship to cellular organisms. For instance, the presence in giant dsDNA viruses, such as mimiviruses, of several genes involved in protein translation, namely for tRNA and tRNA synthetases (but not a single ribosomal protein!), has prompted certain scientists to suggest that these viruses have evolved from cellular organisms by reductive evolution. Our analysis of the large metagenomic dataset definitively demonstrates that genes for different ribosomal proteins have been repeatedly recruited by viruses. Third, we also show that these are not random ribosomal proteins that are selected by viruses but rather those which can be more easily exchanged in the ribosome. Furthermore, our analysis suggests that virus-encoded RPs appear to be differentially selected for across environments as aquatic viruses and those infecting animal-associated bacteria are enriched for different sets of ribosomal proteins. Finally, we provide experimental evidence that phage-encoded ribosomal proteins can be incorporated into bacterial ribosomes and modify their translation properties. Thus, we provide the first global overview on where and what ribosomal proteins are encoded in viral genomes, and some elements pertaining to why some viruses would encode these genes.

For publication in my opinion these hypotheses still require experimental support showing e.g. the alteration of the specificity of protein synthesis triggered by expression of viral encoded ribosomal proteins.

RESPONSE: We agree with the reviewer that demonstrating changes in protein synthesis specificity upon incorporation of a phage-encoded ribosomal protein is an important step, and we have now performed these additional functional tests (revised Fig. 4 and I. 266-278). Specifically, to ensure the homogeneity of ribosomes containing virus-encoded proteins, we tagged the viral and native E. coli bS21 with

streptavidin tag and purified the corresponding ribosomes using the affinity column. The purified ribosomes were then used in the in vitro translation assays. The ribosomes containing the tagged E. coli bS21 efficiently translated the reporter GFP protein, whereas those containing the viral bS21 were much less active (5% activity compared to the wt ribosomes). This indicated that the viral homolog alters the activity of ribosomes. The inactivity of ribosomes carrying the viral bS21 suggests that additional virus- or host-encoded factors may be needed for proper translation, or that the ribosomal protein from a phage infecting Pelagibacter ubique may not be compatible with E. coli ribosomes. Unfortunately, at this point we are not able to perform these experiments in the native Pelagibacter system, where genetic tools are not available and the viruses have not been revivable (Sullivan, unpublished data).

Specific points

Discussion: As mentioned above I miss several publications that provide evidence for the regulation of host protein synthesis by phosphorylation of ribosomal proteins and initiation factors (e.g. Robertson and Nicholson, Biochem. 1992)

RESPONSE: Thank you for pointing these out. We have added the information on protein phosphorylation to the Introduction:

“T7-like podoviruses encode serine/threonine kinases which have been shown to phosphorylate around 90 proteins, including several involved in protein translation, such as ribosomal proteins S1 and S6 as well as translation initiation factors IF1, IF2, and IF3, and elongation factors G and P (Refs 24,25). It was suggested that phosphorylation of these proteins may stimulate translation of the phage late mRNAs.”

Figure 4: Considering the high Mg²⁺ concentration of 10 mM, I am wondering why no peaks representing the ribosomal subunits are visible in the sucrose density gradient! Moreover, the pictures showing the negative staining of the 70S ribosomes do not add any information and could be omitted.

RESPONSE: At high Mg²⁺ concentration, the 70S is very stable and represents a large majority of the purified particles. However, we still can see the peaks corresponding to the 50S and 30S subunits. They are now indicated in the figure. The negative staining EM pictures were moved from the main figure to supplementary information.

In addition, the authors discuss the fact that the HPF is not present in the 70S fraction despite it is expressed and detectable in the cell extract. However, HPF is stabilizing 100S dimers, as the authors correctly mention in the manuscript on page 5, 2nd line! Thus, I would not expect the protein to be present in the 70S fraction!

RESPONSE: Although this was not clear in the original figure, the gradient fractions which were pulled for mass spectrometry analysis contained not only 70S (the major component) but also 100S particles. The dotted lines in Figure 4A indicate which fractions were analyzed. In the revised figure, we labeled the peaks corresponding to the 100S particles, so that a reader can more easily understand which particles were included in each experiment. Thus, our results suggest that virus-encoded HPF was present in neither 70S nor 100S.

Reviewer #2 (Remarks to the Author):

The authors present a study about ribosomal proteins (RPs) in cultivated and uncultivated viruses. As

the authors state certain RPs have been associated with environmental viral sequences previously, however this study went deeper to ascertain their presence in a broader sense. They also show that at least 2 of the 14 proteins appeared to incorporate into the final protein structure when expressed in E. coli. However it is important that function was not shown in this study.

RESPONSE: Although this study primarily focused on the identification of ribosomal protein genes in viral genomes, their diversity and distribution across different environments, we agree with the reviewer that characterizing the function of these genes remains a key question. In the revised manuscript we now include additional functional data on in vitro translation with ribosomes containing the viral bS21 proteins, and show that these can modify the specificity of protein synthesis of E. coli ribosome (please see a more detailed response to reviewer 1).

Line 52: 'take over' having one or two proteins of a complex large system is somewhat misleading, perhaps it is a modulation that is needed and/or there are other unknown imperative function(s) involved in virus/host associations during infection.

RESPONSE: We agree with the reviewer. In the manuscript, we state "Presumably, these viruses are merely tweaking ribosomal functioning in their hosts". In the revised manuscript, we refrained from using "take-over" with regard to ribosomal protein.

Lines 80 -81: Hyphen's as punctuation in a manuscript?

RESPONSE: The hyphens were replaced with em dashes.

Lines 107-108: For the genome size:RP type correlation, can the others discuss this further. Is there any reason why this might be occurring? Was the cutoff arbitrary?

RESPONSE: We thank the reviewer for noting the need for additional discussion on the genome size:RP type correlation. Unfortunately, our observation is made on a relatively small dataset of 15 complete phage genomes encoding ribosomal proteins, therefore it is not possible to draw statistically significant conclusions on the correlation between the genome size and type of RPs. This is due to the fact that the vast majority of identified ribosomal genes are carried by uncultivated phages for which completeness of the genome cannot be verified. We added a sentence stating that "many more phage genomes encoding RP would be needed to verify the significance of this observation" (l. 111-112).

Lines 116-118: "...phylogenetic analysis strongly suggests that the phage gene was horizontally acquired ..." I would like the authors to discuss this more and/or provide more data to support the claim. Based on Figure 1c it can definitely be said that the host and phage reside in the same clade with good support, however it is unclear if the gene went from phage to host or the reverse (the latter suggested by the authors).

RESPONSE: In the phylogeny shown in Figure 1c, bS21 from different families of the class Alphaproteobacteria group together, whereas the mitochondrial homolog occupies the basal position. This placement is most parsimoniously explained by a scenario under which mitochondria have evolved from an alphaproteobacterial ancestor. In this tree, the cellular homologs are basal to the viral sequence, which suggests that gene transfer has occurred from an alphaproteobacterium to a virus. This is now explained in the revised manuscript l. 119-124:

“Maximum likelihood phylogenetic analysis showed that bS21 homologs from different families of alphaproteobacteria cluster together and form a sister group to the mitochondrial homolog, consistent with the scenario under which mitochondria have evolved from an alphaproteobacterial ancestor. In this tree, all alphaproteobacterial sequences are basal to the viral protein, strongly suggesting that the phage gene was horizontally acquired from the *Pelagibacter* host (Figure 1c).”

Lines 125-126: Does this suggest that the protein does not function and was merely an acquisition of genetic material during packaging?

RESPONSE: Although these genes may in theory be not functional, we do not believe it to be the most likely scenario for the following reasons. bL9 protein consists of two globular domains, N-terminal and C-terminal, both of which individually and specifically bind to the 23S rRNA. The N-terminal domain is preserved in the virus-encoded protein, whereas the C-terminal domain is replaced by a non-homologous domain. Thus, in principle, it is possible that the viral protein still binds to the ribosome, but has a distinct function than the native bL9. We have reworded the sentence to read: “While the C-terminal domain in the viral bL9 homolog has been apparently non-homologously replaced with sequence that lacks known function, the N-terminal RNA-binding domain and part of the α -helical spacer are preserved (Figure S2b), suggesting that the viral protein may bind to the 23S rRNA.” (l. 130-133)

Line 161: references Figure S3B. Are the authors correct to reference it here? It is unclear, the legend indicates the arrows point/indicate which RPs were found within viruses but does not specifically state environmental. There is also only 12, but the text says 14 were identified?

RESPONSE: Thank you pointing out this inconsistency. Indeed, S3B is not accurate and we now refer to Figure S3 rather than to panel B only (2 proteins are shown in panel A and 10 in panel B). The legend has been updated and now indicates that “Ribosomal proteins found in viruses and environmental virus contigs are colored and indicated with arrows”. We also state that “The HPF (Ribosomal_S30AE) is not present in the depicted structures”. All 13 identified proteins are shown in Figure 2.

Lines 168-172: I don't see how Figure 3 truly supports the authors claim that there have been “7 distinct virus-host exchanges of bs21”. Please provide more evidence or clarification.

RESPONSE: We acknowledge that the data presented may not be sufficient to fully support the claim of “7 distinct virus-host exchanges of bs21”, and we thank the reviewer for noting this over-statement. Our initial estimation of the number of distinct gene exchange was based on evaluating (i) large monophyletic clades of virus-encoded bs21 separated from other virus sequences by cellular bs21, and (ii) bs21 detected in entirely different genome context, i.e. encoded by unrelated viruses. However, we agree that this suffers from several limitations. For instance, virus-virus gene transfers are rare but not impossible, hence the detection of bs21 in unrelated viruses cannot be strictly considered as a direct evidence for multiple virus-host transfer events. Also, the topology of the bs21 tree includes several long branches and deep-branching virus clades, the placement of which remains uncertain given the limitations of current tree building methods. This is not unusual when trying to build phylogenies spanning across the whole bacterial and archaeal diversity, but this type of deep branches must be cautiously interpreted.

Upon re-examination of this claim, we believe that these limitations preclude a simple and clear assessment of how many exchanges occurred. Nonetheless, the bs21 tree clearly suggests that virus-

encoded bs21 do not derive from a single virus-host exchange event, given how broadly the virus sequences are distributed across the tree. We thus revised the manuscript to remove mentions of a specific number and instead states: “suggested that multiple virus-host exchanges of bs21 protein-coding genes have occurred, likely across various bacterial phyla”.

Line 180: Is the 10 contigs coming from the pool of 78 that had a putative host assigned? Or the total 1,403?

RESPONSE: These 10 contigs are from the total pool. This has now been clarified in Table S2 by adding a column “With host prediction” which provides the numbers of contigs with a putative host assigned for each ribosomal protein.

Lines 188-189: “Thus, at this point, there is an emerging picture that viruses might randomly sample host DNA, including ribosomal protein genes, and that in some cases these might become fixed in viral genomes.” This is not a new idea, but the way written it suggests it.

RESPONSE: We thank the reviewer for pointing out this sentence as confusing. Although the exchange of genes between host and viruses has been known and studied for a long time, ribosomal protein genes have not been previously described and analyzed in viral genomes. The only previous indication that ribosomal genes might be carried by viruses was based on the analysis of short metagenomic contigs from prefiltered “viral fractions”. These short contigs (due to technical limitations of that time) could represent either actual viral genomes or cellular contaminations present in the viromes. It is only now, when longer contigs can be assembled due to improved sequencing and virome preparation protocols, that we can confidently claim that these are actually viral genomes. Furthermore, we show for the first time that ribosomal genes are carried by virus isolates. We have now revised the sentence to clarify as follows: “Thus, at this point, there is an emerging picture that ribosomal protein genes acquired through random sampling of host DNA might, in some cases, become fixed in viral genomes” (l. 195-196)

Lines 202-205: The sentence is repeated.

RESPONSE: Thanks, the repeated sentence has now been deleted.

Lines 221-222: “Thus, phage-encoded bs21 might compete with and replace the cellular bs21, forcing preferential translation of viral transcripts.” I feel like this is a big jump. There is no indication, in the presented data, of preference. However, it does suggest that it gives the virus its own mechanism for translation initiation without relying on host machinery (at least for this one step).

RESPONSE: We have removed “preferential” and reworded the sentence as follows: “Thus, phage-encoded bs21 might compete with and replace the cellular bs21, ensuring translation of viral transcripts.” (l. 229-230).

Lines 231-239: I really do not like the use of pN/pS used for determining or suggesting function. It is used for evolution, yes, but I don’t see the direct link to function. Can the authors clarify why they chose this and also provide any necessary references that show a direct link between this inference and function.

RESPONSE: The estimation of the pN/pS ration is widely used to assess the directionality and magnitude of selection acting on protein-coding genes, e.g. PMID: 23222524, 30223889, 29021524, 29066755, 28085156. However, we agree with the reviewer that this test does not suggest any specific function for

the gene, but rather assess whether a gene is performing any function by estimating the balance between neutral mutations, purifying selection and beneficial mutations. If a gene is randomly acquired from the host genome and does not benefit the virus or is deleterious to the virus, it is expected to evolve under neutral ($pN/pS=1$) or positive ($pN/pS>1$) selection, respectively. The sequences with the pN/pS ratios smaller than 1 are under purifying selection, i.e., their evolution is constrained such that mutations do not alter the protein sequence. Such purifying selection is commonly interpreted to result from selection to maintain function of the encoded protein. We revised the sentence accordingly as follows: “Here we used this metric to test if virus-encoded RPs were under purifying or positive selection, where the former ($pN/pS<1$) would indicate selection for a functional protein and the latter ($pN/pS>1$) would indicate that the gene might be in the process of being phased out from the viral genome⁴⁶. We found that well-sampled viral-encoded RP genes (>10x coverage, and ≥ 1 single nucleotide polymorphism, or SNP) had an average $pN/pS=0.10$, with 84% having a $pN/pS\leq 0.20$ (Table S2). This suggests that these genes are under strong purifying selection and thus likely retained an important function after being transferred into virus genomes.” (l. 239-245).

Lines 278-280: “Nevertheless, genes for the core components of translation machinery, namely RPs, until now appeared as the last unbreached boundary between the cellular and viral kingdoms”. This is confusing since it was stated in the introduction (Line 80-81) that “viral genome fragments ... suggested that viruses might encode ribosomal proteins. Therefore this is not new information for bs21 but it is definitely more in depth analysis.

RESPONSE: While it is true that the existence of virus-encoded ribosomal protein genes had been hypothesized, at the time viral metagenomics studies did not produce long enough contigs to rule out that the sequences found in the viromes came from contaminating host DNA. We have clarified this in the revised introduction: “Curiously, however, viral genome fragments assembled from environmental viral community sequence datasets (viral metagenomes), which vastly expand upon cultured sequence space, suggested that viruses might encode ribosomal proteins, specifically, bS1 and bS21. Though challenges insuring removal of contamination from cellular genomes and the lack of host context available warrants caution about such observations of ‘cellular features’ in metagenome-only datasets (22,28), the findings are intriguing.” (l. 82-86).

Reviewers' Comments:

Reviewer #1:

Remarks to the Author:

In the revised version of the manuscript, 'Numerous cultivated and uncultivated viruses encode ribosomal proteins' the authors introduced changes and added the requested references. In addition, they rephrased the manuscript to clarify their statements. Moreover, they aimed to experimentally support their hypothesis, that phage can reprogram the host translational machinery. However, this functional analysis shown now in Figure 4C does not foster their statement of ribosome heterogeneity and functional specialization. The authors show that *E. coli* ribosomes equipped with pelagiphage-encoded bs21 protein have a strongly reduced translational efficiency. This result is not surprising, as a protein optimized for the *E. coli* ribosome is replaced by a protein encoded by a phage that is specific for a rather distant species. Thus, the additional experiment does not add any information nor strengthens their manuscript. Why don't the authors use a compatible system? E.g. they could use *Salmonella* that is amenable for translational studies and express the ribosomal gene encoded by the *Salmonella* phage and check for translational selectivity comparing mRNAs derived from *Salmonella* and the phage?

Reviewer #2:

Remarks to the Author:

The authors have addressed all comments fully. The inclusion of new experiments to show incorporation of bs21 into the ribosomes added more towards linking the bioinformatic analysis with biology, although there is still more needed to understand the duration of incorporation/use by the phage etc, prevalence, activity in a relevant and tractable bacteria.

One very small observation while looking through the manuscript is that there is not consistency in referring to figures. At times the panel letter is capitalized and others it is not - perhaps this is journal specific, but I point it out in the event it is not.

RESPONSES TO REVIEWERS' COMMENTS:

Reviewer #1 (Remarks to the Author):

In the revised version of the manuscript, 'Numerous cultivated and uncultivated viruses encode ribosomal proteins' the authors introduced changes and added the requested references. In addition, they rephrased the manuscript to clarify their statements. Moreover, they aimed to experimentally support their hypothesis, that phage can reprogram the host translational machinery. However, this functional analysis shown now in Figure 4C does not foster their statement of ribosome heterogeneity and functional specialization. The authors show that *E. coli* ribosomes equipped with pelagiphage-encoded bs21 protein have a strongly reduced translational efficiency. This result is not surprising, as a protein optimized for the *E. coli* ribosome is replaced by a protein encoded by a phage that is specific for a rather distant species. Thus, the additional experiment does not add any information nor strengthens their manuscript. Why don't the authors use a compatible system? E.g. they could use *Salmonella* that is amenable for translational studies and express the ribosomal gene encoded by the *Salmonella* phage and check for translational selectivity comparing mRNAs derived from *Salmonella* and the phage?

RESPONSE: Thank you for taking the time to review our manuscript once again. Unfortunately, homologs of S21 were not identified in known *Salmonella* phages, whereas *Pelagibacter*, which is infected by the S21-encoding phage, is as distant from *Salmonella* as it is from *E. coli*. However, expression of other phage-encoded ribosomal proteins in a broader range of organisms is certainly worth undertaking in the future.

We agree that the additional experiments did not provide definitive proof for the modulation of protein translation by phage-encoded ribosomal proteins. Therefore, we further toned down the conclusions regarding virally encoded RPs modulating host translational machinery (including in the Abstract).

Reviewer #2 (Remarks to the Author):

The authors have addressed all comments fully. The inclusion of new experiments to show incorporation of bs21 into the ribosomes added more towards linking the bioinformatic analysis with biology, although there is still more needed to understand the duration of incorporation/use by the phage etc, prevalence, activity in a relevant and tractable bacteria.

One very small observation while looking through the manuscript is that there is not consistency in referring to figures. At times the panel letter is capitalized and others it is not - perhaps this is journal specific, but I point it out in the event it is not.

RESPONSE: Thank you for taking the time to review our manuscript once again. The manuscript has now been checked for consistency of capitalizations.